# Superposition in Graph Neural Networks

**Lukas Pertl**[*1]    **Han Xuanyuan**[*2]    **Pietro Liò**[1]
[1]University of Cambridge    [2]Independent Researcher
ljfp2@cantab.ac.uk    hxuany@outlook.com    pl219@cam.ac.uk

## Abstract

Interpreting graph neural networks (GNNs) is difficult because message passing mixes signals and internal channels rarely align with human concepts. We study superposition, the sharing of directions by multiple features, directly in the latent space of GNNs. Using controlled experiments with unambiguous graph concepts, we extract features as (i) class-conditional centroids at the graph level and (ii) linear-probe directions at the node level, and then analyze their geometry with simple basis-invariant diagnostics. Across GCN/GIN/GAT we find: increasing width produces a phase pattern in overlap; topology imprints overlap onto node-level features that pooling partially remixes into task-aligned graph axes; sharper pooling increases axis alignment and reduces channel sharing; and shallow models can settle into metastable low-rank embeddings. These results connect representational geometry with concrete design choices (width, pooling, and final-layer activations) and suggest practical approaches for more interpretable GNNs.

## 1 Introduction

Understanding what features a model represents and how they are arranged in latent space is central to trustworthy ML. For graph neural networks (GNNs), this is unusually hard. Unlike pixels or tokens, graphs do not offer a fixed coordinate system across inputs; receptive fields are relational and variable; and many signals are *structural* (motifs, roles, spectral patterns) rather than obviously human readable. As a result, most GNN interpretability work answer *which nodes or edges mattered*, not *what internal features the network formed* or *how those features are arranged* [Ying et al., 2019, Luo et al., 2020, Yuan et al., 2022, Kakkad et al., 2023].

A key obstacle to interpretability is *superposition*: many features are packed in fewer directions, creating polysemantic channels and entangled geometries [Elhage et al., 2022, Scherlis et al., 2023]. Superposition has been studied in MLPs and transformers, but its behavior in GNNs, where message passing and pooling constrain geometry, remains underexplored. This motivates the central theme of our work, which asks: *How does superposition arise in GNNs, how do architectural and graph-structural choices modulate it, and what are the downstream consequences for interpretability?*

To answer these questions, we build small, controllable datasets where the relevant graph concepts are unambiguous (e.g., "two adjacent identical types" or "contains a triangle and a hexagon"), train standard GNNs, and look directly at the geometry of their internal representations – both at the node level (before pooling) and at the graph level (after pooling). We then use two simple, basis-invariant diagnostics: one that says "how many distinct axes are effectively used" and another that says "how tightly packed the directions are compared with an ideal arrangement."

Our study yields several main findings:

- **Width produces a phase pattern.** As the final hidden layer widens, feature overlap first decreases, then briefly increases around the point where capacity matches the number of con-

---

[*]Equal contribution.

cepts, then decreases again. At high width, graph-level features approach near-ideal packing, but node-level concepts often remain entangled – evidence that readout layers can disentangle what the message-passing stack keeps mixed.

- **Pooling sharpness encourages axis alignment.** Making pooling more "winner-take-all" (e.g., max pooling) increases alignment to coordinate axes and reduces node-level feature sharing. We explain this with two complementary arguments: gradients concentrate on large coordinates, and under noise the axis-aligned choice loses less information than an oblique one.

- **Rank collapse can appear even in shallow GNNs.** In some runs the numerical rank of the pooled representation stays below the number of learned features while the accuracy remains high. We link this to hard gating after aggregation (exactly zeroing channels) and to a loss-driven preference for "mutually obtuse" class directions that resists activating new dimensions.

**Contributions.** (i) A representation-centric framework for analyzing superposition in GNNs that works across architectures and layers; (ii) robust, simple diagnostics and feature extraction procedures (centroids and probes) that avoid assumptions about neurons or coordinates; (iii) empirical maps of how width, topology, and pooling shape superposition; and (iv) practical guidance – when to expect entanglement, when pooling helps, and how to avoid low-rank traps.

## 2 Background

### 2.1 Superposition and the interpretability of internal representations

Classic unit analyses in CNNs found channels that act like concept detectors [Zeiler and Fergus, 2014, Zhou et al., 2014, Bau et al., 2017, Olah et al., 2017]; related work spans RNNs and transformers [Karpathy et al., 2015, Strobelt et al., 2017, Gurnee et al., 2023, 2024, Geva et al., 2020], and early GNN studies exist [Xuanyuan et al., 2023]. However, concepts need not be axis-aligned: they can correspond to arbitrary directions or nonlinear detectors. Directional tools (TCAV, linear probes) therefore treat a feature as a vector in latent space [Kim et al., 2018, Alain and Bengio, 2016]; recent sparse-autoencoder work aims to demix polysemantic units [Bricken et al., 2023, Gao et al., 2024].

*Superposition* is the phenomenon where models represent more features than available dimensions by allowing features to share neurons/directions, producing *polysemantic* units and entangled directions [Elhage et al., 2022]. Both neuron-level and mechanistic analyses benefit from (approximate) decomposability: having independently meaningful, linearly separable features. Superposition violates this assumption, obscuring neuron-level semantics and complicating circuit reconstruction. Methods that demix features (e.g., sparse autoencoders) can mitigate these issues [Bricken et al., 2023, Gao et al., 2024].

### 2.2 Representations in GNNs

A message-passing layer aggregates neighbor information and applies a local transform,

$$\mathbf{H}^{(l+1)} = \phi\Big(\text{AGG}(\mathbf{H}^{(l)}, \mathbf{A}), \mathbf{W}^{(l)}\Big), \tag{1}$$

with architecture-specific aggregation (e.g., normalized sum for GCN [Kipf and Welling, 2017], unnormalized sum + MLP for GIN [Xu et al., 2019], attention for GAT Veličković et al. [2018]). After $L$ layers, node embeddings $\mathbf{H}^{(L)}$ are pooled into a graph embedding $\mathbf{h}_G = g(\mathbf{H}^{(L)})$ via sum/mean/max.

**Superposition pressure in GNNs.** Message passing repeatedly *mixes* many neighbor features into fixed-width hidden states. This creates capacity pressure that encourages *superposition* – multiple features sharing the same channels – leading to polysemantic units and entangled directions. While superposition has been analyzed in toy feedforward/transformer settings, to our knowledge there is little analogous work for GNNs. This gap further complicates neuron-level semantics and the reconstruction of GNN circuits.

# 3 Quantifying Superposition in Representations

## 3.1 Interpreting and extracting features

Prior work often reasons about superposition at the *input* to layers (e.g., pixels/tokens). For GNNs, we study it *where it matters* for interpretability: in the latent spaces of nodes and graphs. We ask: *how many features are packed into how few directions, and how well are those directions separated?*

Following [Alain and Bengio, 2016, Kim et al., 2018, Elhage et al., 2022, Bricken et al., 2023, Gao et al., 2024], we treat a *feature* as a direction in a model's latent space that supports a prediction. For a GNN with forward pass

$$\mathbf{X} \xrightarrow{\phi_1} \mathbf{H}^{(1)} \xrightarrow{\phi_2} \cdots \xrightarrow{\phi_L} \mathbf{H}^{(L)} \xrightarrow{g} \mathbf{h}_G \to \hat{\mathbf{y}},$$

each $\phi_l$ is a message-passing block, $\mathbf{H}^{(l)} \in \mathbb{R}^{n_l \times d_l}$ are node embeddings, $g$ is a graph-pool, and $\mathbf{h}_G \in \mathbb{R}^d$ is the graph embedding. We interpret features in two complementary ways.

**(1) Linear-probe features (model-decodable concepts).** Linear probes are a well established method for finding feature directions [Akhondzadeh et al., 2023]. Given embeddings $\{\mathbf{h}_G\}$ (graph-level) or rows of $\mathbf{H}^{(l)}$ (node-level) and binary targets $y_\ell \in \{0, 1\}$ for concept $\ell$, we fit a logistic probe on a held-out split:

$$s_\ell = \mathbf{w}_\ell^\top \mathbf{z} + b_\ell, \quad \mathbf{z} \in \{\mathbf{h}_G\} \text{ or } \{\mathbf{H}_{i:}^{(l)}\}.$$

The *probe normal* $\mathbf{w}_\ell$ (unit-normalized) is taken as the feature direction for concept $\ell$. Since all our geometry is directional, the intercept $b_\ell$ is irrelevant.

**(2) Class-conditional centroids (task-aligned concepts).** On a held-out split, we form one-hot sets using the ground truth $\mathbf{y}$.

$$\mathcal{S}_\ell^{\mathrm{GT}} = \{G : \mathbf{y}(G) = \mathbf{e}_\ell\}, \qquad \mathbf{c}_\ell = \frac{1}{|\mathcal{S}_\ell^{\mathrm{GT}}|} \sum_{G \in \mathcal{S}_\ell^{\mathrm{GT}}} \mathbf{h}_G,$$

and stack rows to obtain $C \in \mathbb{R}^{K \times d}$. In the linear/LDA regime, discriminants align with class means [Alain and Bengio, 2016], so centroid directions approximate task-discriminative axes and aggregate the combined effects of message passing and pooling.

**Active features.** Not all labels are reliably learned in every seed, thus including non-existent or highly confused features would distort superposition measurements. We therefore define an *active* feature $\ell$ using only the held-out split:

- Probes: a concept $\ell$ is *active* if $\mathrm{AUC}_\ell \geq 0.60$.
- Centroids: class $\ell$ is *active* if in-class recall $\geq 0.5$ and all off-diagonal recalls $< 0.5$.

Let $\mathcal{L}_{\mathrm{act}}$ be the active set ($|\mathcal{L}_{\mathrm{act}}| = k_a$). We form the *feature matrix* by stacking unit vectors: either $\{\hat{\mathbf{w}}_\ell\}_{\ell \in \mathcal{L}_{\mathrm{act}}}$ (probes) or $\{\hat{\mathbf{c}}_\ell\}_{\ell \in \mathcal{L}_{\mathrm{act}}}$ (centroids). In the main results we use class-conditional centroids plus linear probes as complementary views: centroids assess task-aligned geometry, probes assess model-decodable directions.

## 3.2 Proposed Metrics

We propose metrics to measure superposition for a $k_a \times d$ feature matrix $C$.

**Effective rank (EffRank).** With singular values $\sigma_1 \geq \sigma_2 \geq \cdots$ and $p_i = \sigma_i / \sum_j \sigma_j$, the entropy-based effective rank is

$$\mathrm{EffRank}(C) = \exp\left(-\sum_i p_i \log p_i\right). \tag{2}$$

It estimates the number of *effectively used* axes. For **centroids** we COM-center $C^\circ = C - \mathbf{1}\bar{\mathbf{c}}^\top$ with $\bar{\mathbf{c}} = \frac{1}{k_a} \sum_\ell \mathbf{c}_\ell$ before computing EffRank, removing a global bias direction that no practitioner would interpret as a mechanism. For **probe normals** we use the raw $C$.

**Superposition Index (SI).** To express features per effective axis we use the absolute index

$$\mathrm{SI} = \frac{k_a}{\mathrm{EffRank}(C)}. \tag{3}$$

$\mathrm{SI} = 1$ indicates no extra sharing beyond having $k_a$ independent directions; $\mathrm{SI} > 1$ means multiple features share axes (greater superposition). This avoids dependence on the ambient $d$.

**Welch-Normalized Overlap (WNO).** While SI measures the mean number of features distributed over each axis, it does not consider their angular geometry[2]. Therefore to complement SI, we introduce an angular overlap metric called the *Welch-normalized overlap*: let $\widetilde{C}$ be the matrix obtained from $C$ by normalizing each row to unit $\ell_2$ norm and define

$$\overline{\cos^2} = \frac{2}{k_a(k_a - 1)} \sum_{i < j} \langle \tilde{\mathbf{c}}_i, \tilde{\mathbf{c}}_j \rangle^2.$$

We compare $\overline{\cos^2}$ to (i) the random baseline $1/d_{\mathrm{eff}}$ (independent unit vectors in $\mathbb{R}^{d_{\mathrm{eff}}}$) and (ii) the Welch lower bound $\mu_*^2 = \max\left(0, \frac{k_a - d_{\mathrm{eff}}}{d_{\mathrm{eff}}(k_a - 1)}\right)$, and report

$$\mathrm{WNO} = 1 - \frac{\frac{1}{d_{\mathrm{eff}}} - \overline{\cos^2}}{\frac{1}{d_{\mathrm{eff}}} - \mu_*^2}, \tag{4}$$

so $\mathrm{WNO} = 0$ is Welch-optimal (least overlap), $\mathrm{WNO} = 1$ is random, and $\mathrm{WNO} > 1$ is worse-than-random packing. Since ambient $d$ can be much larger than the subspace actually used. We therefore report *intrinsic* WNO: set $r = \lceil \mathrm{EffRank}(C^\dagger) \rceil$ (with $C^\dagger = C^\circ$ for centroids and $C^\dagger = C$ for probes), project $C$ to the top-$r$ right-singular subspace to obtain $C_r$, row-normalize, and take $d_{\mathrm{eff}} = r$ in (4). For **centroids** and **graph-level probes** we additionally remove a single shared offset direction before the SVD (PC1 removal); for **node-level probes** we do *not* remove PC1.[3]

**Invariances and scope.** EffRank and WNO are invariant to right-multiplication by orthogonal matrices (basis changes) and to uniform rescaling of rows; they probe geometry intrinsic to the representation rather than coordinate choices. SI quantifies features-per-axis pressure; WNO quantifies how tightly those axes are packed relative to principled baselines.

## 4 Effects of Model Width and Graph Topology

### 4.1 Datasets

**The PAIRWISE dataset.** We introduce a dataset where graphs are equal length linear chains of 20 nodes. Each node has an $n$-dimensional one-hot/zero feature. Class $k$ is active if two adjacent nodes both activate dimension $k$, yielding sparse multi-class targets. The task is attribute driven: since each class is associated with identical graph structure, we can probe for superposition whilst factoring out the effects of varying topology. Sparsity is controlled by the activation probability $p$ of a node having a one-hot (non-zero) feature. Exact details for reproduction are included in Appendix A.1.

**The CONJUNCTION dataset.** To isolate *topology–driven* superposition from effects of extraneous input attributes, we introduce a graph family in which all node features are initialized to the node's degree, and supervision otherwise depends only on the presence of simple cycles. Let $C_\ell(G)$ denote the indicator that graph $G$ contains at least one simple cycle of length $\ell$. We instantiate four motif detectors $\ell \in \{3, 4, 5, 6\}$ (triangles, squares, pentagons, hexagons) and define two labels by *conjunctions* of cycles:

$$y_A(G) = \mathbf{1}\{ C_3(G) \wedge C_6(G) \}, \qquad y_B(G) = \mathbf{1}\{ C_4(G) \wedge C_5(G) \}.$$

All other graphs receive $y_A = y_B = 0$. Thus the same lower-level *motif concepts* $\{C_3, C_4, C_5, C_6\}$ are re-used across the two tasks, but each label requires a distinct pair. This cleanly separates (i)

---

[2]For example, two different arrangements of packing 6 vectors into 2D: (i) arranged as a near-regular hexagon, and (ii) arranged such that there are two antipodal clusters. Both have EffRank $\approx 2$.

[3]PC1 removal is appropriate when a single global bias dominates all classes (e.g., pooled magnitude); removing it improves sensitivity to relative packing. For node probes this shared offset is not guaranteed and we keep the raw geometry. If $r \leq 1$ or $k_a \leq 1$, WNO is undefined and we report `NA`.

the formation of topological features from (ii) how a model combines them. A graph is formed by sampling required counts for the four motifs according to the label pattern and then adding $\mathrm{Binomial}(8, 0.5)$ extra copies of each motif. Exact details for reproduction are included in Appendix A.2. Motif instances are placed on disjoint node sets and connected by random bridges to avoid trivial overlaps. Because there are no informative node attributes, any successful solution must *construct* the motif detectors purely via message passing.

## 4.2 Width-induced superposition

We study how width $d$ affects superposition on PAIRWISE ($k = 16$) on 16 seeds. The first hidden layer is set to the input width ($k$) and the second layer to $d$, inducing a bottleneck when $d < k$. We measure (a) class-conditional *centroids* in the graph space and (b) node-level concepts $\mathsf{Is}_t$ and $\mathsf{NextTo}_t$ via linear probes on the last pre-pooling layer: let $x_i \in \{e_1, \ldots, e_k\}$ be the one-hot node type and $\mathcal{N}_i$ the neighbors of $i$. For type $t$,

$$\mathsf{Is}_t(i) := \mathbf{1}\{x_i = e_t\}, \qquad \mathsf{NextTo}_t(i) := \mathbf{1}\{\exists j \in \mathcal{N}_i : x_j = e_t\}.$$

We keep only *active* features: centroids with in-class recall $> 0.5$ and off-class recall $< 0.5$, and probes with AUROC $> 0.6$. We report *intrinsic* WNO (computed in the EffRank-selected subspace). Centroids are COM-centered for EffRank; probe normals are not. Optimizer and training details for this experiment and subsequent ones are given in B.1.

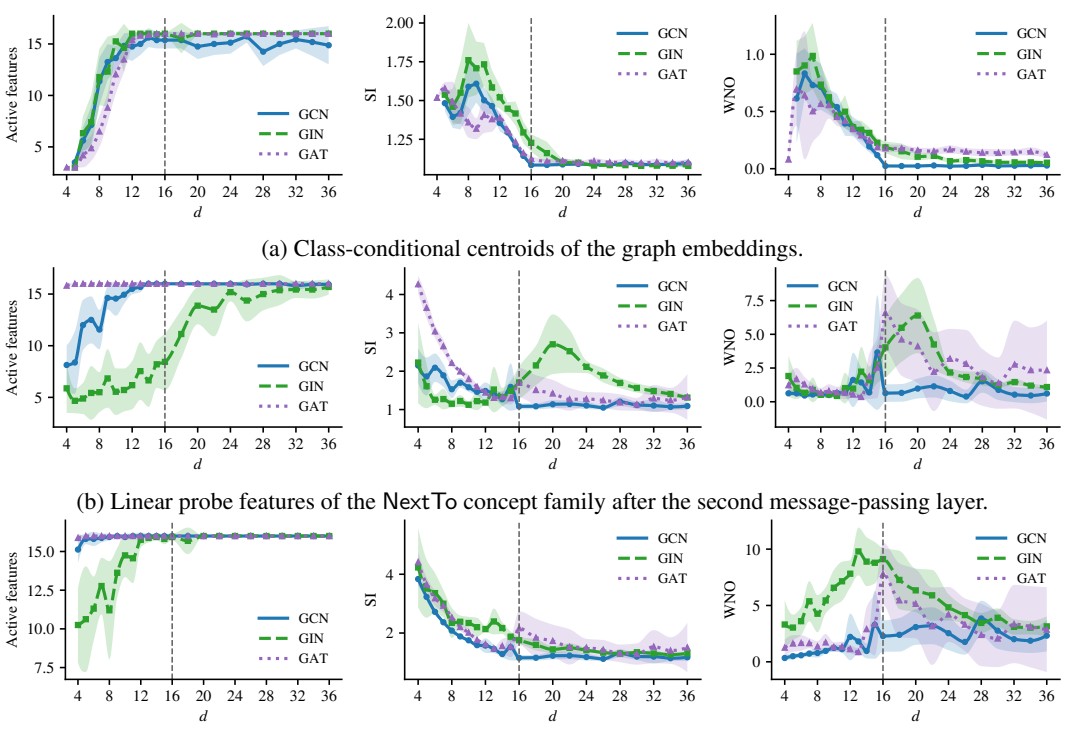

(a) Class-conditional centroids of the graph embeddings.

(b) Linear probe features of the NextTo concept family after the second message-passing layer.

(c) Linear probe features of the Is concept family after the second message-passing layer.

Figure 1: Superposition effects across bottleneck dimensions $d$. A dashed line corresponds to $k = d$. Shaded regions show uncertainty as mean $\pm 1.96\sigma/\sqrt{R}$ where $R = 16$ is the number of seeds.

We observe a consistent *three-phase* transition with increasing $d$ for both graph centroids and node concepts (GCN/GIN/GAT): an initial improvement in packing (SI↓, WNO↓), a reversal near $d \approx k$ (SI↑, WNO↑), and a final decay back to good packing. The effect is stronger for node features: centroids approach Welch-optimal geometry (WNO→ 0, SI→ 1), whereas node probes often stabilize around random or worse packing (WNO $\geq$ 1). This suggests that superposition is largely *resolved by the readout* but not eliminated at the last node layer.

Architecture trends follow inductive bias. For GIN, the post-aggregation MLP tends to funnel many labels into shared axes ("generic neighbor/type pressure + small residuals"), which the readout later

separates by sign/magnitude at the graph level. GAT shows peaks that are present but typically smaller; attention can separate channels earlier, while GCN produces smoother, less volatile curves.

The WNO/SI peak typically coincides with saturation of the number of active features/probes and the onset of a flat test loss. This aligns with two competing effects of more width: *feature formation* (easier to instantiate additional task-useful directions) versus *packing efficiency* (easier to spread existing directions). Early on, packing dominates; around $d \approx k$ capacity is spent on forming new directions (SI/WNO rise); once formation saturates, packing dominates again and SI/WNO drop. Centroids are tied directly to the BCE objective, so their feature formation saturates earlier, explaining the shorter first phase.

### 4.3 Topology-induced superposition

On CONJUNCTION we train 3-layer GCN, GIN, and GATv2 models (16 channels per layer), apply a global `mean pool` followed by a ReLU, and fit a two-logit readout $(y_A, y_B)$. For each architecture we repeat training over 250 seeds. We use probes to study two feature families: (i) *node-level* $\mathsf{Inside}_\ell(i)$ on the last pre-pooling layer, which activates if node $i$ is part of an $\ell$-cycle, and (ii) *graph-level* $\mathsf{Has}_\ell(G)$ on the post-pooling representation, which activates if the graph $G$ contains any $\ell$-cycle.

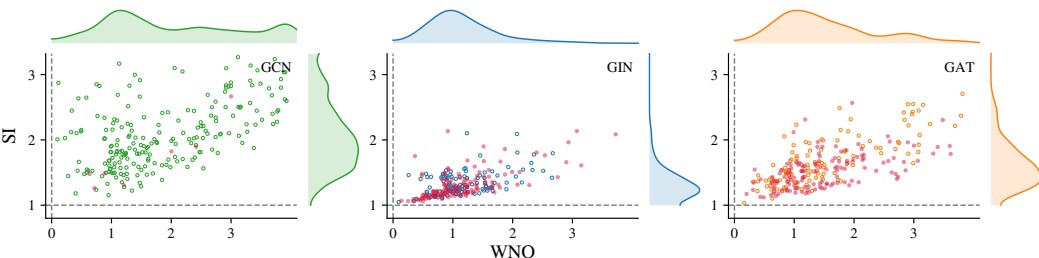

Figure 2: SI and WNO measured on GNNs trained on CONJUNCTION. Red highlighted data-points indicate models with perfect test accuracy.

**Despite $d \gg k$, superposition remains large.** With $k=4$ motifs and width $d=16$, independent axes would yield SI $\approx 1$ and WNO $\approx 0$. Instead, Fig. 2 shows clusters at SI $> 1$ and WNO $\gg 0$, i.e., above random overlap. In most runs, at least one motif pair has $|\cos| > 0.9$ (Table 1), most often $(C_3, C_4)$ or $(C_5, C_6)$; in rare cases ($\sim$1% in GIN) all four are nearly collinear, consistent with an emergent lever of 'cycleness'. High accuracy can still be achieved, suggesting that the pooling / reading takes advantage of residual multicoordinate structure or magnitude even when node-level directions coalesce.

**Cosine geometry reveals a *cyclic* structure.** Figure 3 reports mean cosine similarity matrices. At the *node* level, the lengths of the nearby cycles align (for example, $C_3$ with $C_4$, $C_5$ with $C_6$) while distant pairs are weakly aligned or slightly antialigned. From the standard spectral view, a depth-$L$ message-passing GNN implements a low-degree polynomial filter in the normalized adjacency matrix $\tilde{\mathbf{A}}$ [Defferrard et al., 2016, Kipf and Welling, 2017]: ignoring nonlinearities one can write

$$\mathbf{H}^{(L)} \approx \sum_{k=0}^{L} \tilde{\mathbf{A}}^k \mathbf{X} \Theta_k,$$

where $\mathbf{X} \in \mathbb{R}^{n \times d_{\text{in}}}$ are the input node features, $\mathbf{H}^{(L)} \in \mathbb{R}^{n \times d_{\text{out}}}$ are the layer-$L$ node embeddings, and each $\Theta_k \in \mathbb{R}^{d_{\text{in}} \times d_{\text{out}}}$ is a learnable coefficient matrix specifying how information arriving via $k$-step walks is mixed across feature channels. The power $\tilde{\mathbf{A}}^k$ encodes $k$-step walks on the graph, and its diagonal entries $(\tilde{\mathbf{A}}^k)_{ii}$ correspond to closed walks of length $k$ at node $i$. Detectors for $C_3, \ldots, C_6$ therefore necessarily reuse many of the same short closed-walk monomials, which naturally makes nearby cycle lengths more aligned in representation space. After *pooling*, geometry re-mixes toward the task: the post-pool ReLU and linear readout learn graph-level axes that are linear combinations of node-level detectors (e.g., in GIN with mean-pool, $C_3$ aligns with $C_6$ while $C_4$ opposes $C_5$, which helps reject single-motif false positives).

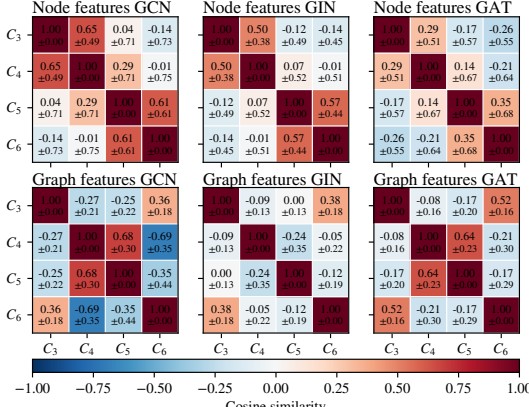

Figure 3: Mean cosine similarities for the node features (top) and graph features (bottom) on CONJUNCTION.

Table 1: Percentage of instances above 0.9 absolute cosine similarity.

| Model | Pool | Node all | Node one | Graph all | Graph one |
|---|---|---|---|---|---|
| GCN | mean | 15.1% | 87.8% | 0.0% | 44.9% |
| GCN | max | 6.7% | 78.8% | 0.0% | 20.2% |
| GIN | mean | 1.1% | 44.4% | 0.0% | 1.1% |
| GIN | max | 0.4% | 41.2% | 0.0% | 9.2% |
| GAT | mean | 2.4% | 64.1% | 0.0% | 10.0% |
| GAT | max | 0.7% | 63.0% | 0.0% | 12.2% |

**Max pooling reduces lever sharing.** Since a linear probe's sign can flip with an equivalent boundary, we track $|\cos|$ to detect shared levers. Switching from mean to max pooling reduces node-level alignment across all architectures: both the fraction of runs with any $|\cos| > 0.9$ pair and with all pairs $> 0.9$ drop (Table 1). Intuitively, a per-channel max cannot encode a conjunction on a single channel – one large activation would mask the other—so the model is pressured to place motifs on distinct channels, lowering $|\cos|$. The ordering in Table 1 (GCN > GAT > GIN in lever sharing) aligns with each layer's mixing bias. GCN's degree normalized sum acts as a low-pass smoother, concentrating signals into a few graph-harmonic axes and driving different motifs to share directions. GAT's learned attention reduces indiscriminate mixing and yields moderate de-alignment, but a single head offers limited selectivity. GIN's unnormalized sum plus post aggregation MLP provides the most expressive channel rotation and gating, producing more specialized directions and the lowest node and graph level alignment.

**Discussion.** Even with generous width, semantically different topology features occupy *similar* node-space directions. Max pooling alleviates but does not eliminate this, probably reflecting message-passing inductive biases. Note that many high-accuracy models sit at SI > 1 and WNO ≫ 0 (Fig. 2), indicating that accurate solutions often rely on node-level lever sharing with disentanglement deferred to the graph-level readout – superposition here is not merely a symptom of under-capacity.

## 5 Pooling-Induced Axis Alignment of Graph Representations

**Metric and preprocessing.** We measure axis preference with the *Alignment Index* (AI) computed on any set of feature directions $\{c_\ell \in \mathbb{R}^d\}_{\ell=1}^k$ (node-level probe normals or graph-level class features):

$$\text{AI} = \frac{1}{k} \sum_{\ell=1}^{k} \frac{\max_j |(c_\ell)_j|}{\|c_\ell\|_2}.$$

Thus, $\text{AI} \approx 1/\sqrt{d}$ for random orientations and $\text{AI} \to 1$ when each feature is concentrated on a coordinate axis. We always use row-unit directions; for *centroids* we COM-center across classes before normalizing (to remove the shared offset), while *probes* use the raw fitted normals.

**Global and local pooling conditions.** For the graph readout we use the signed power-mean family (details in Appendix C) that interpolates mean ($p = 1$) and max ($p \to \infty$):

$$y_d = \text{sgn}(s_d) N^{-1/p} |s_d|^{1/p}, \qquad s_d = \sum_{i=1}^{N} \text{sgn}(x_{id}) |x_{id}|^p.$$

To isolate *local* effects, we also replace the last message-passing aggregator by the same power-mean operator while holding the global readout at mean pooling.

**Alignment vs. pooling and regime.** On PAIRWISE ($k = 16$) across both a bottleneck ($d = 10$) and a wide ($d = 22$) regime, AI increases as $p > 1$ (Fig. 4). With *global* pooling the effect is clearest at the

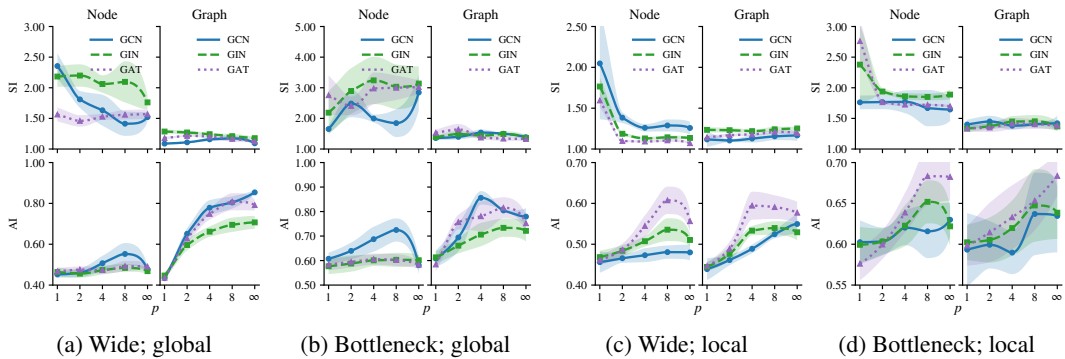

| (a) Wide; global | (b) Bottleneck; global | (c) Wide; local | (d) Bottleneck; local |

Figure 4: Alignment versus pooling parameter $p$ across architectures on PAIRWISE ($k = 16$). Shaded regions show uncertainty as mean $\pm 1.96\sigma/\sqrt{R}$ where $R = 16$ is the number of seeds.

graph level (the nonlinearity acts after node mixing). With a *local* power-mean in the last layer, AI increases at both node and graph levels because the winner-take-most bias precedes the final ReLU and readout. Occasionally AI dips for $p > 2$ in the bottleneck regime; this coincides with less stable training and noisier features. The trends persist when conditioning on high-accuracy runs. GATv2 tends to exhibit higher node-level AI even for modest $p_{\text{local}}$ (attention already induces axis-selective flows), whereas GCN's degree-normalized linear aggregation produces smoother trends.

**From alignment to constrained superposition.** Under mean pooling, strong bottlenecks yield large SI due to shared directions, especially at the *node* level where separation is not directly optimized by the loss. Increasing $p_{\text{local}}$ reliably *reduces* $\text{SI}_{\text{node}}$ (Fig. 4), effectively capping features-per-axis. Raising $p_{\text{global}}$ also lowers $\text{SI}_{\text{node}}$ in the wide regime (consistent with §4.3); in the bottleneck regime the dimensional constraint dominates and this effect largely vanishes. $\text{SI}_{\text{graph}}$ stays near-flat: in the wide regime it is already close to optimal under mean pooling (§4.2), and in the bottleneck regime additional alignment cannot overcome the shortage of dimensions.

**Noise as the driver of axis alignment.** Two complementary arguments explain why increasing $p > 1$ breaks rotational symmetry in favor of axis alignment. *(i) Learning-dynamics view.* Backpropagating through generalized mean pooling (App. C) yields, for each pooled feature $h_d$,

$$\frac{\partial h_d}{\partial x_{id}} = \frac{1}{N}\big(|\bar{s}_d| + \varepsilon\big)^{\frac{1}{p}-1}\big(|x_{id}| + \varepsilon\big)^{p-1},$$

(a derivation is given in Appendix D) where $\bar{s}_d$ is the mean of the transformed features and $\varepsilon > 0$ is the stabilizer. For fixed $d$ and $p$, the prefactor $\frac{1}{N}\big(|\bar{s}_d| + \varepsilon\big)^{\frac{1}{p}-1}$ is shared across nodes, so the *relative* gradient magnitudes are controlled by $(|x_{id}| + \varepsilon)^{p-1}$: with $p = 1$ we recover $\partial h_d/\partial x_{id} = 1/N$ (a rotation-invariant Jacobian), whereas for $p > 1$ coordinates with larger $|x_{id}|$ receive disproportionately larger updates, steadily aligning features to coordinate axes. In the absence of noise all orientations become equivalent once the model has recognized the features; in realistic graphs ubiquitous irrelevant features inject noise and trigger the symmetry breaking. *(ii) Geometric view.* Under equal-energy corruption, axis-aligned vectors lose less information than randomly oriented ones; in high dimensions the components of a random unit vector scale like $1/\sqrt{d}$ and are easily swamped by noise spikes. Figure 5 visualizes this effect under max pooling. We numerically simulate the equal-energy corruption under max pooling: for each $(n, \sigma)$ on a grid (dimensions $n = 2 \ldots 20$, noise levels $\sigma \in [0.001, 0.35]$) we sample 100 random vectors $v \in \mathbb{R}^n$ (either uniform on the unit sphere or one-hot), draw 20 i.i.d. Gaussian noise candidates per coordinate with variance $\sigma^2$, and replace each coordinate by the candidate with largest absolute value whenever its magnitude exceeds $|v_i|$.

**Practical takeaways.** For more axis-aligned graph features, prefer global pooling with $p_{\text{global}} > 1$. To make node channels less polysemantic, consider a power-mean local aggregator in the final layer.

## 6 Rank collapse and metastable superposition

Our geometry metrics (SI/WNO) quantify *how* features pack *inside the span* of the learned representations. They do not say *how many* dimensions that span actually occupies. Let $H \in \mathbb{R}^{N \times d}$ stack

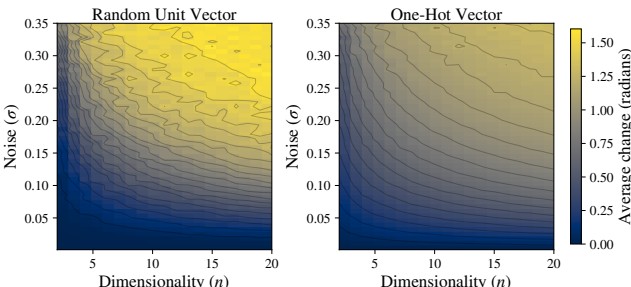

Figure 5: Axes-aligned embedding vectors lose less information than arbitrary-angled vectors under max pooling when both are corrupted by equal-energy noise.

pooled graph embeddings (rows are graphs) and let $C \in \mathbb{R}^{k_a \times d}$ stack the $k_a$ active class centroids. Since centroids are linear averages of rows of $H$, $C = A^\top H R$ for suitable averaging/projection matrices, hence

$$\text{EffRank}(C) \ \leq \ \text{rank}(C) \ \leq \ \text{rank}(H).$$

If the span of $H$ is low-dimensional, *superposition is inevitable*: by pigeonhole, $k_a > \text{rank}(H)$ forces multiple features to share axes. We formalize this with a numerical rank

$$r_\tau(H) \ = \ \#\{i : \sigma_i(H)/\sigma_1(H) \geq \tau\},$$

and call a run *collapsed* when $r_\tau(H) < k_a$.[4] This is the regime where high SI/WNO are not just a matter of suboptimal packing but a *hard capacity constraint* from a thin span.

**Two training modes.** On PAIRWISE we observe two characteristic dynamics (Fig. 6). In some seeds $r_\tau(H)$ rises to the layer width ($d = 16$), with $\text{EffRank}(H)$ tracking from below; SI/WNO on centroids drop accordingly. In other seeds $r_\tau(H)$ remains strictly below $d$, sometimes far below $k_a$: SI/WNO stay high and the smallest singular values show brief activation attempts (spikes) that recede, indicating the optimizer returns to a low-rank basin. Interestingly, across models that exhibit overfitting (such that the right model in Fig. 6) we observe sharp multi-phase transitions in SI and WNO throughout training. A possible (non-rigorous) explanation is provided in Appendix E.

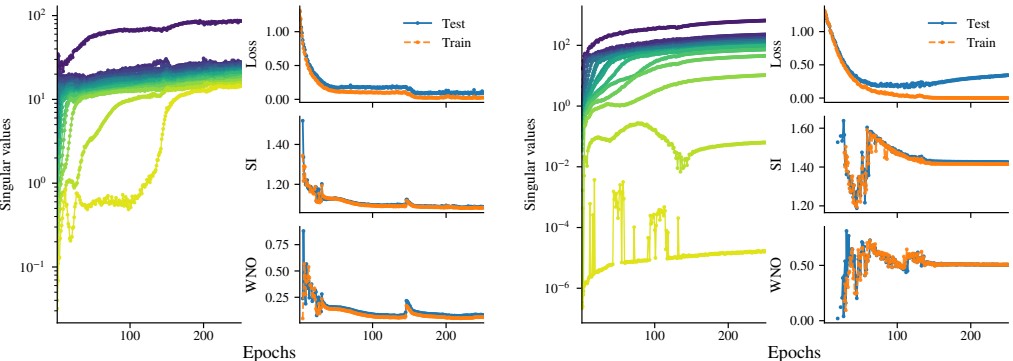

Figure 6: Examples showing the evolution of singular values of a GCN model (left) and GIN model (right).

**Why does the low-rank basin persist?** Two effects make it metastable: (i) *Global channel gating by ReLU*. If a channel is negative across the dataset just before the last nonlinearity, ReLU zeros it everywhere, so the corresponding column of $H$ vanishes and algebraic rank drops exactly. Replacing the final ReLU with LeakyReLU substantially reduces such dead columns (App. F.5). (ii) *Obtuseness pressure under BCE*. Before a separating hyperplane forms, last-layer directions $\{\mathbf{v}_i\}$ tend to align with their class weights $\{\mathbf{w}_i\}$ and become mutually obtuse. Moving one feature into a fresh orthogonal

---

[4]We use $\tau = 10^{-4}$; results are qualitatively unchanged for $\tau \in [10^{-5}, 10^{-3}]$. An energy criterion $r_\eta(H) = \min\{r : \sum_{i \leq r} \sigma_i^2 \geq (1 - \eta) \sum_i \sigma_i^2\}$ gives the same conclusions.

dimension rotates it towards $90°$ against the others and increases the BCE loss. After the hyperplane forms, the cross-class margins satisfy $\mathbf{w}_j^\top \mathbf{v}_i < 0$ $(i \neq j)$, and perturbing $\mathbf{v}_k$ into a new dimension makes those dots less negative, again increasing loss (Appendix F). Hence gradients oppose escapes until the accuracy gain from a new dimension outweighs the obtuseness penalty, explaining the singular jump events.

**When is collapse likely?**

We see more collapse (i) with narrower last layers, (ii) in GIN (post-aggregation ReLUs) than GCN (no ReLU after the final aggregation), and (iii) as the number of task features grows. This agrees with the regime analysis in Appendix F: when $k_a \leq d + 1$, mutually obtuse low-rank configurations exist and can act as local minima; as $k_a$ approaches or exceeds $2d$, such configurations become geometrically impossible and $r_\tau(H)$ tends to reach $d$.

**Connection to oversmoothing.** Classic oversmoothing is within-graph: node states become indistinguishable with depth. Here we see a cross-graph analogue: pooled graph embeddings concentrate in a thin global subspace. Accurate models can then separate classes inside that span by sign/magnitude, which explains why SI/WNO can be high at the *node* level yet acceptable in the *graph* space.

**Discussion.** The width-sweeps in §4.2 showed SI/WNO can be high even when $d \geq k$; here we identify a complementary mechanism: training can settle into *thin-span* solutions, so overlap is *forced* (because $k_a > r_\tau(H)$) rather than just suboptimal packing within a rich span. This also clarifies why node-level superposition can persist while graph-level centroids look well-packed: the readout operates *inside* a low-dimensional subspace, separating classes by sign/magnitude without increasing the span of $H$.

# 7   Conclusion

In this work, we provided, to the best of our knowledge, the first systematic study of superposition in GNNs. Concretely, we introduced a representation-centric framework that quantifies feature sharing in GNNs via class-conditional centroids and linear-probe directions. Across architectures, width induces a three-phase trajectory in overlap; topology shapes node-level geometry that pooling re-mixes toward the task; and sharper pooling drives axis alignment and reduces lever sharing. We also observed metastable low-rank graph embeddings in shallow models. Practically, modest increases in width, LeakyReLU in the final MLP, and sharper but stable pooling improve interpretability without sacrificing accuracy. A key next step is to link superposition dynamics to over-smoothing and over-squashing, integrating our geometric view with spectral, dynamical, and topological analyses.

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

# A  Datasets

## A.1  The PAIRWISE dataset

**Graph structure and node features.**  The PAIRWISE dataset consists of graphs that are simple chains of fixed length $L = 20$. We represent the chain as an undirected path on nodes $\{1, \ldots, L\}$ with edges $(i, i + 1)$ for $i = 1, \ldots, L - 1$, encoded as two directed edges $(i, i + 1)$ and $(i + 1, i)$; in our experiments we do *not* add self-loops. Each node $i$ carries a $k$-dimensional feature vector $x_i \in \{0, 1\}^k$, where $k$ is the number of categories (we denote the corresponding dataset by PAIRWISE$_k$). Node features are sampled independently as follows: first draw an activation flag $b_i \sim \mathrm{Bernoulli}(p)$ with $p = 0.9$; if $b_i = 1$, sample a category $c_i \sim \mathrm{Unif}\{1, \ldots, k\}$ and set $(x_i)_{c_i} = 1$ and all other entries to 0, while if $b_i = 0$ we set $x_i = \mathbf{0}$. Thus each node is either all-zero or one-hot, with a high expected fraction $p$ of active nodes. All random choices are made with a fixed seed (42) for reproducibility.

**Label definition, deduplication, and split.**  Given a feature matrix $X \in \{0, 1\}^{L \times k}$ on the chain, the multi-label target $y \in \{0, 1\}^k$ is defined coordinate-wise by

$$y_c = \mathbf{1}\Big\{ \exists\, (i, j) \in E \text{ with } (x_i)_c = (x_j)_c = 1 \Big\}, \qquad c = 1, \ldots, k,$$

where $E$ is the set of chain edges (self-loops, if present, are ignored when computing $y$). In words, class $c$ is active if and only if there exists at least one edge whose two endpoints both carry the one-hot category $c$. To build the dataset, we repeatedly sample node features $X$ as above, compute $y$, and retain only unique graphs, where uniqueness is defined by the node-feature pattern: we hash $X$ (after casting to `uint8`) and reject duplicates until we have $n_{\mathrm{train}} + n_{\mathrm{test}} = 3000$ distinct samples. We then randomly shuffle these 3000 graphs (using the same seed 42) and split them into $n_{\mathrm{train}} = 2000$ training and $n_{\mathrm{test}} = 1000$ test graphs, corresponding to a test ratio of $1/3$.

## A.2  The CONJUNCTION dataset

**Data generation.**  We instantiate the CONJUNCTION dataset by drawing $n_{\mathrm{train}} = 3000$ and $n_{\mathrm{test}} = 1000$ graphs with a fixed random seed of 42. For each graph we first sample a latent category $c \in \{\text{none}, A\text{-only}, B\text{-only}, \text{both}\}$ from a categorical prior $(0.25, 0.25, 0.25, 0.25)$. Conditioned on $c$, we specify the *required* counts $r_\ell \in \{0, 1\}$ of cycles of lengths $\ell \in \{3, 4, 5, 6\}$ as follows. For $A$-only we set $(r_3, r_4, r_5, r_6) = (1, 0, 0, 1)$; for $B$-only $(0, 1, 1, 0)$; for both $(1, 1, 1, 1)$. For the "none" category we sample uniformly from the four non-target pairs $(1, 0, 1, 0)$, $(0, 1, 0, 1)$, $(1, 1, 0, 0)$, $(0, 0, 1, 1)$, which guarantees that neither conjunction $C_3 \wedge C_6$ nor $C_4 \wedge C_5$ holds. Given these required counts, we then draw independent extra copies $e_\ell \sim \mathrm{Binomial}(8, 0.5)$ for each length $\ell$ with $r_\ell = 1$, and set the total motif counts to $n_\ell = r_\ell + e_\ell$; lengths with $r_\ell = 0$ remain absent. This corresponds to a symmetric Bernoulli prior over up to eight additional copies per present motif. The final labels are then computed deterministically from the counts as

$$y_A(G) = \mathbf{1}\{n_3 \geq 1 \wedge n_6 \geq 1\}, \qquad y_B(G) = \mathbf{1}\{n_4 \geq 1 \wedge n_5 \geq 1\},$$

so that the four categories correspond to $(y_A, y_B) \in \{(0, 0), (1, 0), (0, 1), (1, 1)\}$ as intended.

**Graph construction.**  Given the counts $(n_3, n_4, n_5, n_6)$, we materialize a concrete graph by placing all motif instances on disjoint node sets. For each of the $n_\ell$ copies of a cycle of length $\ell$, we create a simple cycle on $\ell$ new nodes and designate one "anchor" node on that cycle. To increase local structural variety while preserving the core motifs, we attach "whisker" paths to each anchor: for every anchor we sample a number of whiskers $W \sim \mathrm{Unif}\{0, 1, 2\}$, and for each whisker we sample a length $L_{\mathrm{wh}} \sim \mathrm{Unif}\{1, 2\}$ and attach a simple path of length $L_{\mathrm{wh}}$ starting from the anchor. Letting $\mathcal{A}$ denote the set of anchors of all motif instances, we then randomly permute $\mathcal{A}$ and connect consecutive anchors by simple paths of length $L_{\mathrm{br}} \sim \mathrm{Unif}\{1, 2\}$, ensuring that the overall graph is connected while the cycles themselves remain node-disjoint. Node features are one-dimensional and equal to the node degree (i.e., we use purely structural degree features with no learned or random attributes).

**Deduplication and split.**  To avoid distributional artefacts from re-using the same graph topology, we perform bucketed rejection sampling with Weisfeiler–Lehman (WL) deduplication. We maintain four buckets (one per category) and generate candidate graphs as above until each bucket contains the desired number of graphs implied by the class prior (subject to rounding), rejecting any candidate

whose 1-WL hash (obtained from three iterations of 1-WL colour refinement on the undirected graph and hashing the resulting colour histogram) has already been seen in that bucket, with a cap of 60,000 proposals per bucket. Once a total of 4000 unique graphs have been collected, we perform a per-bucket random split into 3000 training and 1000 test graphs using the same seed (42), which preserves the class proportions up to integer rounding.

# B  Optimizer and training

## B.1  Optimizer details

All models are trained with the Adam optimizer as implemented in PyTorch, using the default parameters $(\beta_1, \beta_2) = (0.9, 0.999)$ and $\epsilon = 10^{-8}$. Unless otherwise stated, we use a mini-batch size of 256, no learning-rate scheduling, and no early stopping; every configuration is run over multiple random seeds, but the optimizer hyperparameters are identical across seeds. No dropout is performed.

Table 2: Training hyperparameters (optimizer, learning rate, weight decay, and batch size) by experiment and architecture.

| Experiment | Task / dataset | Model | Learning rate | Weight decay | Epochs |
|---|---|---|---|---|---|
| 1 | PAIRWISE$_{16}$ | GCN | 0.10 | 0 | 300 |
| 1 | PAIRWISE$_{16}$ | GIN | 0.01 | 0 | 150 |
| 1 | PAIRWISE$_{16}$ | GAT | 0.01 | 0 | 150 |
| 2 | CONJUNCTION | GCN | 0.001 | $10^{-5}$ | 1600 |
| 2 | CONJUNCTION | GIN | 0.001 | $10^{-5}$ | 1600 |
| 2 | CONJUNCTION | GAT | 0.001 | $10^{-5}$ | 1600 |
| 3 | PAIRWISE$_{16}$ | GCN | 0.10 | 0 | 200 |
| 3 | PAIRWISE$_{16}$ | GIN | 0.01 | 0 | 200 |
| 3 | PAIRWISE$_{16}$ | GAT | 0.01 | 0 | 200 |
| 4 | PAIRWISE$_{16}$ | GCN | 0.10 | 0 | 250 |
| 4 | PAIRWISE$_{16}$ | GIN | 0.01 | 0 | 250 |
| 4 | PAIRWISE$_{16}$ | GAT | 0.01 | 0 | 250 |

**Experiment 1 (Dimension induced bottleneck on PAIRWISE).**   We use mean global pooling, batch size 256, and the learning rates and weight decays from Table 2. We train GCN models for 300 epochs and GIN/GAT models for 150 epochs:

GCN: lr $= 0.10$, wd $= 0$, 300 epochs;     GIN/GAT: lr $= 0.01$, wd $= 0$, 150 epochs.

The optimizer is Adam with the above defaults and no scheduler.

**Experiment 2 (Topology induced superposition).**   For the second experiment we again use Adam, batch size 256, and mean pooling, but employ smaller learning rates and a small amount of weight decay. All architectures share the same training schedule:

GCN/GIN/GAT: lr $= 10^{-3}$, wd $= 10^{-5}$, 1600 epochs.

Training is performed by calling `model.fit(train_loader, optimizer, num_epochs=1600)` with no additional regularization beyond weight decay.

**Experiment 3 (alignment on PAIRWISE$_{16}$).**   In the alignment experiments on the PAIRWISE$_{16}$ dataset we sweep the hidden dimension over $[10, 16, 22]$ with batch size 256, and the same Adam learning rates and weight decays as in Experiment 1 (see Table 2). We sweep the generalized-mean pooling exponent over

$$p \in \{1.0, 2.0, 4.0, 8.0, p_{\max}\},$$

where $p_{\max}$ denotes the largest exponent used in our code (approximating max pooling); all other optimization hyperparameters are kept fixed across $p$.

**Experiment 4 (rank / selectivity on $\textsc{Pairwise}_{16}$).** We use the `ExperimentRank` wrapper with hidden dimension 16, batch size 256, and the Adam settings from Table 2. Each model is trained via `model.fit(train_loader, optimizer, num_epochs=num_epochs)`, where `num_epochs` is a fixed constant within this experiment and does not depend on the random seed; no learning-rate schedule or additional regularization is applied beyond the specified weight decay.

## C  Generalized Mean Pooling

Let $N$ be the number of nodes, $x_{id} \in \mathbb{R}$ the $d$-th dimension of node $i$, and $p \in \mathbb{R}$ a pooling parameter. Introduce a small stabiliser $\varepsilon > 0$ and define

$$g_{p,\varepsilon}(x) \;=\; \mathrm{sgn}(x)\big(|x| + \varepsilon\big)^p.$$

**1. Element-wise transformation:** For every node $i$ and feature dimension $d$,

$$\tilde{x}_{id} = g_{p,\varepsilon}(x_{id}) = \mathrm{sgn}(x_{id})\big(|x_{id}| + \varepsilon\big)^p.$$

**2. Pooling:** Sum the transformed features and take their mean,

$$s_d \;=\; \sum_{i=1}^N \tilde{x}_{id}, \qquad \bar{s}_d \;=\; \frac{1}{N}\, s_d.$$

**3. Inverse transformation:** Apply the signed $p$-th root to obtain the pooled feature

$$h_d \;=\; \mathrm{sgn}(\bar{s}_d)\big(|\bar{s}_d| + \varepsilon\big)^{1/p}.$$

The resulting graph-level representation is the vector

$$\mathbf{h} = \big(h_1, h_2, \ldots, h_D\big).$$

*Remark.* Setting $p = 1$ recovers ordinary mean pooling (modulo the stabilisation term $\varepsilon$), while $p \to \infty$ recovers max pooling.

## D  Symmetry Breaking

Recall the stabilized generalized mean pooling from App. C. For each feature dimension $d$, let $x_{id} \in \mathbb{R}$ be the feature of node $i$, and define

$$\tilde{x}_{id} = g_{p,\varepsilon}(x_{id}) = \mathrm{sgn}(x_{id})\big(|x_{id}| + \varepsilon\big)^p, \qquad s_d = \sum_{i=1}^N \tilde{x}_{id}, \qquad \bar{s}_d = \frac{1}{N} s_d.$$

The pooled feature is then

$$h_d = \mathrm{sgn}(\bar{s}_d)\big(|\bar{s}_d| + \varepsilon\big)^{1/p}.$$

We now compute $\partial h_d / \partial x_{id}$, away from the measure-zero points where $x_{id} = 0$ or $\bar{s}_d = 0$ (so that the sign functions are locally constant).

**Step 1: derivative of the inner transform.** For fixed $d$ and node $i$, write

$$\tilde{x}_{id} = \mathrm{sgn}(x_{id})\big(|x_{id}| + \varepsilon\big)^p.$$

Using $\frac{\partial |x_{id}|}{\partial x_{id}} = \mathrm{sgn}(x_{id})$ and $\mathrm{sgn}(x_{id})^2 = 1$, we obtain

$$\frac{\partial \tilde{x}_{id}}{\partial x_{id}} = \mathrm{sgn}(x_{id}) \cdot p\big(|x_{id}| + \varepsilon\big)^{p-1} \cdot \frac{\partial |x_{id}|}{\partial x_{id}} = p\big(|x_{id}| + \varepsilon\big)^{p-1}.$$

Hence

$$\frac{\partial s_d}{\partial x_{id}} = \frac{\partial}{\partial x_{id}}\Big(\sum_{j=1}^N \tilde{x}_{jd}\Big) = \frac{\partial \tilde{x}_{id}}{\partial x_{id}} = p\big(|x_{id}| + \varepsilon\big)^{p-1},$$

and therefore

$$\frac{\partial \bar{s}_d}{\partial x_{id}} = \frac{1}{N}\frac{\partial s_d}{\partial x_{id}} = \frac{p}{N}\big(|x_{id}| + \varepsilon\big)^{p-1}.$$

**Step 2: derivative of the outer map.** Define $y = \bar{s}_d$ and write

$$h_d = f(y) = \operatorname{sgn}(y)\big(|y| + \varepsilon\big)^{1/p}.$$

For $y \neq 0$, $\operatorname{sgn}(y)$ is locally constant, and we have

$$\frac{\partial h_d}{\partial y} = \operatorname{sgn}(y) \cdot \frac{1}{p}\big(|y| + \varepsilon\big)^{1/p-1} \cdot \frac{\partial |y|}{\partial y} = \frac{1}{p}\big(|y| + \varepsilon\big)^{1/p-1},$$

since $\frac{\partial |y|}{\partial y} = \operatorname{sgn}(y)$ and the sign factors cancel.

**Step 3: chain rule.** Combining the two steps by the chain rule,

$$\frac{\partial h_d}{\partial x_{id}} = \frac{\partial h_d}{\partial \bar{s}_d} \cdot \frac{\partial \bar{s}_d}{\partial x_{id}} = \frac{1}{p}\big(|\bar{s}_d| + \varepsilon\big)^{1/p-1} \cdot \frac{p}{N}\big(|x_{id}| + \varepsilon\big)^{p-1}.$$

Thus the stabilized gradient of generalized mean pooling is

$$\boxed{\frac{\partial h_d}{\partial x_{id}} = \frac{1}{N}\big(|\bar{s}_d| + \varepsilon\big)^{\frac{1}{p}-1}\big(|x_{id}| + \varepsilon\big)^{p-1}}$$

for all $x_{id}$ and $\bar{s}_d$ away from the non-differentiable points of the absolute-value function.

*Remark.* If we further let $\varepsilon \to 0$ and assume non-negative activations so that $\bar{s}_d = s_d/N$ and $x_{id} \geq 0$, this reduces to $\frac{\partial h_d}{\partial x_{id}} = N^{-1/p}|s_d|^{1/p-1}x_{id}^{p-1}$.

# E   Sharp SI transitions during transition

**A training-time phase transition.** On PAIRWISE we observe two training modes: (i) *full-rank* runs where $r_\tau(H)$ rises to $d$ and $\operatorname{EffRank}(H)$ tracks it; (ii) *low-rank* runs where $r_\tau(H)$ remains $< d$ for most of training. Across seeds, SI/WNO of the *centroids* exhibit a reproducible four-stage pattern:

1. **Feature birth (noisy).** Early on, new class directions appear; $\operatorname{EffRank}(C)$ rises $\Rightarrow$ SI↓. Packing inside the span is crude, so WNO typically *increases*.

2. **Compression (sharp transition).** A narrow span forms: smallest singular values of $H$ drop or remain suppressed, PC1 energy rises, and $\operatorname{EffRank}(C)$ contracts $\Rightarrow$ a *sharp SI spike*. This point almost always coincides with the onset of overfitting (test loss starts to rise).

3. **Repacking (smooth).** One or two small singular directions "wake up", allowing repacking *within* a slightly larger span: WNO↓ and SI↓ smoothly.

4. **Convergence (flat).** Geometry stabilizes; train loss drifts down, test loss drifts up.

The sharpness of Stage 2 reflects a discrete change in $r_\tau(H)$ (entire channels becoming globally inactive or reactivated), whereas WNO is computed *after* PC1 removal and need not jump at the same epoch.

# F   Geometric Regimes for Anti-Aligned Embedding Directions

## F.1   Problem Formulation

We first consider the case where the feature number $n$ is small and no hyperplane has formed: Let $\big\{\mathbf{v}_i \in \mathbb{R}^d \mid \|\mathbf{v}_i\|_2 = 1,\ i = 1, \ldots, n\big\}$ be the (column-normalized) embedding directions learned by the network's last hidden layer, and let $\{\mathbf{w}_i\}_{i=1}^n$ denote the corresponding unit-length rows of the classifier weight matrix. The anti-alignment hypothesis we explored in Section 6 states that after training

$$\mathbf{v}_i \approx \mathbf{w}_i \qquad \text{and} \qquad \mathbf{v}_i^\top \mathbf{v}_j \leq 0 \quad (i \neq j), \tag{A.1}$$

i.e. every embedding is (almost) identical to its class weight, and pairwise inner products are non-positive if possible. In other words, the network attempts to realise a set of $n$ mutually obtuse vectors in $\mathbb{R}^d$. The feasibility of (A.1) depends on the relation between $n$ and $d$, which can be separated into four distinct regimes.

### F.2 Regime Analysis

1. **Over-complete case,** $n > 2d$**.** Via a simple counting argument, at least two vectors must lie in the same closed half-space; hence their inner product is $> 0$, so one pair is necessarily acute.

2. **Intermediate regime,** $d + 1 < n < 2d$**.** There are at most $n = d + 1$ mutually obtuse vectors. In this regime some vectors are mutually orthogonal and some are mutually obtuse.

3. **Simplex threshold,** $n = d + 1$**.** The largest set of vectors that can satisfy $\mathbf{v}_i^\top \mathbf{v}_j < 0$ for all $i \neq j$ is the vertex set of a centred regular $d$-simplex. This is the first value of $n$ that permits pairwise obtuse directions.

4. **Under-complete case,** $n \leq d$**.** One can choose any $n$ orthogonal unit vectors (or the $n$ vertices of a degenerate $n$-simplex in a $n - 1$ dimensional subspace).

### F.3 Implication for Rank Collapse

Assume training has converged to a low-rank subspace $\mathcal{S} \subset \mathbb{R}^d$ with $\dim(\mathcal{S}) = r < d$ and all $\mathbf{v}_i \in \mathcal{S}$. Activating a new latent dimension $\mathbf{e}_\perp \perp \mathcal{S}$ perturbs one vector as $\mathbf{v}_k \mapsto \tilde{\mathbf{v}}_k = \sqrt{1 - \varepsilon^2}\, \mathbf{v}_k + \varepsilon \mathbf{e}_\perp$. Then, for $i \neq k$,

$$\tilde{\mathbf{v}}_k^\top \mathbf{v}_i = \sqrt{1 - \varepsilon^2}\, \mathbf{v}_k^\top \mathbf{v}_i \; \geq \; \mathbf{v}_k^\top \mathbf{v}_i, \tag{A.3}$$

because $\mathbf{v}_k^\top \mathbf{v}_i \leq 0$ in the metastable state. Hence the angle decreases[5] towards $90°$, which reduces pairwise obtuseness. As discussed in Section 6 this is is penalised by the BCE loss function. During back propagation gradients oppose the escape from the low-rank basin, effectively creating a metastable minima. If $\varepsilon$ grows large enough performance gains may begin to outweigh the angle penalty, and the rank grows.

### F.4 Rank Collapse with Hyperplane

So far we have assumed that

$$\mathbf{v}_i \; \approx \; \mathbf{w}_i,$$

yet when a separating hyperplane emerges, the learned embeddings $\{\mathbf{v}_i\}$ and the classifier weights $\{\mathbf{w}_i\}$ are no longer almost identical. Empirically, $\mathbf{v}_i$ are often acute rather than mutually obtuse, but the *cross-class* dot products stay strictly negative:

$$\mathbf{w}_j^\top \mathbf{v}_i \; < \; 0 \qquad (i \neq j). \tag{A.4}$$

**Perturbation into a fresh dimension.** Using the same notation as before, we perturb one embedding as

$$\tilde{\mathbf{v}}_k \; = \; \sqrt{1 - \varepsilon^2}\, \mathbf{v}_k + \varepsilon \, \mathbf{e}_\perp.$$

For any $j \neq k$

$$\mathbf{w}_j^\top \tilde{\mathbf{v}}_k \; = \; \sqrt{1 - \varepsilon^2}\, \mathbf{w}_j^\top \mathbf{v}_k + \varepsilon \, \mathbf{w}_j^\top \mathbf{e}_\perp \; > \; \mathbf{w}_j^\top \mathbf{v}_k, \tag{A.5}$$

since $\mathbf{w}_j^\top \mathbf{v}_k < 0$ by (A.4) while $\mathbf{w}_j^\top \mathbf{e}_\perp = 0$. Thus the cross-class dot product becomes *less* negative, which increases BCE loss. In short, even after the hyperplane forms, the network remains trapped in a low-rank metastable basin for the same reason as before: cross-class obtuseness ($\mathbf{w}_j^\top \mathbf{v}_i < 0$) resists escapes that would reduce the margin.

### F.5 Leaky ReLU Rank

Figure 7 illustrates the evolution of the rank of pooled embeddings across 15 training runs of a GIN model on the PAIRWISE dataset. The left graph presents results for a standard GIN, while the right graph shows outcomes when the MLP layers in the GIN architecture use leaky ReLU as the activation function.

---

[5]Equality holds only when $\mathbf{v}_k^\top \mathbf{v}_i = 0$, keeping the angle unchanged at $90°$.

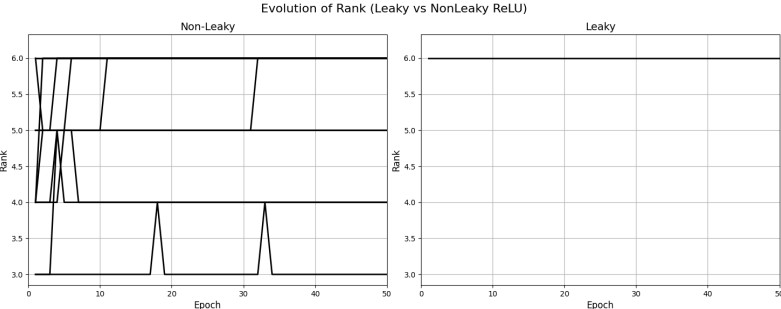

Figure 7: GIN model training runs with input dimension 12 and hidden dimensions [12, 6].

