# OpenReview forum: "Superposition in Graph Neural Networks"
_NeurIPS.cc/2025/Workshop/UniReps — UniReps2025_

### Official Review · Reviewer_q9Gu · 2025-09-04
**Initially a very exciting contribution, but strongly recommending further revisions after a careful reading**

**Confidence:** 5

**Review:**

**[Summary]**

This work aims to propose a representation framework to analyze superposition in GNNs. Node‑level directions (linear probes) and graph‑level class centroids (COM‑centered) are extracted, and feature packing is evaluated using effective rank, SI, and WNO. In superposition disentanglement experiments, the authors argue that (i) last‑layer width induces a three‑phase pattern in overlap; (ii) topology drives node‑level entanglement that pooling remixes into task‑aligned graph axes; (iii) sharper pooling promotes axis alignment and reduces channel sharing, with a plausible gradient/noise explanation; (iv) shallow models exhibit metastable low‑rank spans where superposition is forced.

**[Strengths]**

- Representation‑level methodology (centroids + probes) are applicable across architectures.

- Some metrics/diagnostics are basis-invariant, which is a desideratum for interpretability in this setting.

- The assserted three-phase behavior with respect to width is very intriguing, assuming it is true.

**[Weaknesses]**

- I write about this below.

**[Originality]**

It is a good contribution, assuming it is well-justified. Studying superposition geometry in GNNs is timely, and to my knowledge, underdeveloped.

**[Clarity]**

I rate this poorly, with many missing details and unclear descriptions. I write more about this below.

**[Soundness]**

Due to significant missing details, I do not believe this can be rigorously assessed in the empirical results. The metrics/variants used are reasonably standard, but they should be properly cited, which the paper does not do at all.

**[Quality of Results]**

Despite it's initial impressive appearance, I now judge this to be poor due to the issues detailed below. Additionally, the only datasets evaluated are fully synthetic. Critical details such as dataset size, dataset hyperparameters, etc, are missing, and there are significant concerns with reproducibility.

**[Significance]**

This paper's significance could be good, but only when assuming the results are well-justified. However, I would argue that they are not.

**[General Review]**

This paper is difficult to read, not because the concepts introduced are complex, but because they seem to be presented almost intentionally in an unclear way with many missing bits of information. Conceptually, the paper is not complicated. However, I strongly encourage the authors to consider making the presentation more clear, as readers are likely to spend a significant amount of time trying to guess out what the authors meant (which is inefficient), which forces most readers to quickly skim through the paper unless they have a genuine interest in engaging with the material.

Generally, I ask the authors to define concepts before using them (especially for ones that the authors internally defined and used for evaluations), or at least point to where it can be found.

**[Questions]**

Q1) In lines 181-182, what is Inside()? What is Has()?

Q2) In line 91-92, what is $\mathbf{y}$? How is it computed or defined?

Q3) In line 116, what is "Row-unit–normalized projection"? Is this even a projection? It sounds like the normalization step breaks the projection. If this is the case, the word projection should not be used as it already has meaning.

Q4) Line 132 says that in the PAIRWISE dataset, "graphs are equal length linear chains". But the length is never given in the paper.

Q5) The paper does not provide critical details necessary for generating the datasets used. What are the values of $p$ and $p_{extra}$ for PAIRWISE and CONJUNCTION? Lines 138-149 does not give enough details to be able to reproduce CONJUNCTION even with $p_{extra}$. I suggest the authors make their code public. Details like exact chain length, distribution over graph sizes, or the bridge sampling scheme are not fully specified. This includes the synthetic dataset size--depending on what this value is, I may be significantly less likely to consider the empirical results to be substantive.

Q6) What about results on non-synthetic/artificially generated datasets? The paper does not evaluate on real-world data at all, even rudimentary ones.

Q7) In Figure 1, it is unclear what the shaded region means--is this a confidence interval, or is this min/max over different random seeds? There is no discussion on this.

Q8) How many random seeds were used in Section 4.2? The seed count is given for 4.3, but not 4.2.

Q9) What are the hyperparameters used for the optimizer? Would this not be relevant? Also, what optimizer was even used for the paper? I cannot locate this information.

Q10) If metrics or facts have been published before, it is good practice to cite them instead of stating without justification. For instance, what is "MPNN layer L implements a low-degree polynomial in the normalized adjacency" (lines 192-193) supposed to mean? Can the authors make this more precise? It also does not trivially follow without significant (yet completely missing) justification that "so detectors for $C_3,...C_6$ reuse monomials of closed walks up to length $\approx 2L+1$".

Q11) Lines 171–172, “WNO/SI peak coincides with saturation of active features/probes and onset of flat test loss”. Where is this justified? There are no plots or evaluations showing the test loss at all. I'm unwilling to accept this assertion without justification, as the saturation of active features correlating with onset of flat test loss for graph neural nets is a substantive claim.

Q12) What dataset is Figure 3 evaluated on? This information is missing. What about Figure 4? Or Figure 5? Or Table 1, for that matter? Figure 2 specifies that the networks are trained on CONJUNCTION. Can you clarify this for Figure 3 and Table 1 as well, as I assume it is for CONJUNCTION? As for Figure 4/5, I genuinely have no idea what datasets have been evaluated. It should be possible to generate Figure 5 without a dataset, by sampling random unit vectors. In such cases, I very strongly suggest explicitly including details on the empirical setup, instead of simply writing "Figure 5 (from our diagnostic)" in line 250.

Q13) In Appendix A/B, why take out the stabilization term in the differentiation? This should be explicitly justified or stated, as currently, the differentiation is incorrect. The correct derivative is (note that the sign terms cancel):

$$ \frac{\partial h_d}{\partial x_{i d}}=\frac{1}{N}\left(\left|\bar{s}_d \right|+\varepsilon\right)^{\frac{1}{p}-1}\left(\left|x_{i d}\right|+\varepsilon\right)^{p-1}. $$

Note: The markdown does not render correctly for some reason.

Q14) The paper mixes up the conventions between $v_i^{\top} v_j \leq 0$ and $v_i^{\top} v_j < 0$. In the former case (which the paper explicitly states is used), one does not get the counting argument in Appendix E.2. In the latter case, one does. I encourage the authors to restrict to the latter case for precision.

**[Final Recommendation]**

I recommend a rejection, or acceptance only after a significant amount of revisions. In the current state, the paper has significant concerns for reproducibility due to unclear or missing details, and is generally unnecessarily difficult to read.

**Score:**

1

**Topic Fit:**

3

---

### Official Review · Reviewer_sgs4 · 2025-09-14
**Superposition in Graph Neural Networks**

**Confidence:** 3

**Review:**

This paper claims that GNNs are difficult to interpret correctly, since of message passing
mixes signals and internal channels rarely align with human concepts. To ease the problem, the paper suggests superposing the sharing of directions by multiple features, directly in the latent space of GNNs.

The paper claimed to be the first to study systematic superposition in GNNs. and provide many graphs which are essential for justifies the method.
I like the general idea of the paper, but
1. The graphs need more explanations.
2. No data is provided regarding the input; how the input graphs are generated w/o that, it is hard to appreciate the provided documents
3. A leading example could help the readability of the results

**Score:**

2

**Topic Fit:**

2

---

### Official Review · Reviewer_xqCL · 2025-09-16

**Confidence:** 3

**Review:**

**Summary:** This paper presents a systematic study of superposition in different kind of graph neural networks (GIN, GAT, GCN), a topic that is currently under-investigated yet crucial for understanding GNN representations. Through controlled experiments on two proposed synthetic datasets, the authors investigate how architectural choices like model width and pooling sharpness influence the geometry of learned node- and graph-level features.

**Strengths:**

- The topic of superposition in graphs is an interesting, under investigated one. Understanding how features are entangled in GNNs is a crucial for demystifying the internal dynamics of the message passing mechanism, and this paper makes a foundational contribution.
- The work connects abstract representational geometry to concrete architectural choices, providing practical guidance for designing more interpretable GNNs.

**Weaknesses:**

While the paper is interesting, there are a few areas where it could be improved. I hope the authors find the following suggestions helpful.

1. The paper's positioning could be strengthened by engaging with more of the recent literature on linear probes for graphs. For instance, [1] also use linear probing to analyze the information captured in GNN embeddings similarly to this paper. Situating this in relation to the presented work would provide valuable context and better underscore the novelty of its contribution.
2. The paper's use of synthetic datasets is a clear strength for controlled analysis. However, a key question is how these findings generalize to real-world graphs where features do not encode structural properties of the graph, e.g. in molecules features are properties of atoms and it would be interesting to see if representations encode structural properties of the molecule as well. The paper's impact could be significantly broadened by including even a small-scale experiment on a real-world dataset to demonstrate that the observed phenomena (like the width-induced phase pattern or the effect of pooling) hold in more complex settings.
3. The paper might benefit from an improvement in its clarity in some parts:

    3.1. In paragraph 167-170, authors mention that "Architecture trends follow inductive bias", followed by the actual trends for each GNN. However, the link between architectural inductive biases and the resulting geometric effects could be explained in more details. For example, providing more intuition for why GIN’s MLP funnels features into shared axes.

    3.2. Figure 1 is missing a reference in the main text, making the conclusions drawn in section 4.2 difficult to follow.

    3.3. In line 184, the paper refers to "clusters" of models with high superposition in Figure 2, but the data points appear more as a diffuse distribution. Clarifying what defines these clusters would help the reader better interpret the results.

**Minors and typos:**

- In the diagram of the GNN forward pass (line 82-83), the pooling function $g$ appears to be misplaced. According to the text, pooling maps the final node embeddings $\mathbf{H}^{(L)}$ to the graph embedding $\mathbf{h}_G$. So, the arrow from $\mathbf{H}^{(L)}$ to $\mathbf{h}_G$ should likely be labeled with $g$.
- The notation $\ell$ in line 86 is used to index both the concept and the GNN layer. Using different symbols would avoid potential confusion.
- The appendices provide valuable additional details, but not all of them appear to be referenced in the main text. Ensuring all appendices are referenced where relevant would be helpful for reading the paper.

[1] Akhondzadeh et al., Probing Graph Representations

**Score:**

3

**Topic Fit:**

3

---

### Official Review · Reviewer_KpBp · 2025-09-16

**Confidence:** 5

**Review:**

## Summary

This paper provides a systematic study of superposition in graph neural networks (GNNs). While superposition has been studied in MLPs and transformers, its behavior in GNNs is underexplored. The authors design controlled datasets to tease out superpostion (PAIRWISE: local adjacent pairs of one-hot features and CONJUNCTION: graphs formed from cycles with labels defined by motif conjunctions). They train standard GNNs (GCN, GIN, GAT), then extract features as graph-level class centroids and node level linear probe directions, and analyze their geometry using simple, basis-invariant diagnostics: effective rank (EffRank), superposition index (SI), and Welch normalized overlap (WNO). The authors study the impact of width, topology, and pooling.

## Strengths

* **Clarity**: The paper is well written and a pleasure to read.
* **Novelty**: One of the very first systematic studies of superposition in GNNs.
* **Contribution**: Provides a clean methodology to analyze the impact of design choices on superposition. The work suggests practical heuristics for improving interpretability.
* **Relevance**: The paper is relevant to UniReps, and, more broadly, to the graph learning community.

## Weaknesses

* **Synthetic scope**: Results are on controlled datasets and tasks. It remains open whether the same patterns hold in real-world graph datasets and tasks.
* **Theory**: The explanations are mostly heuristic and descriptive. A more formal theoretical account would strengthen the contribution.

## Suggestions (forward-looking)

* Extend the analysis to real-world datasets and tasks.
* Develop a more formal theoretical model of the observed phenomena.
* Explore whether reducing superposition  directly improves interpretability in practice.

**Score:**

5

**Topic Fit:**

3